# Epidemiology, management and outcomes of *Cryptococcus gattii* infections: A 22-year cohort

**Jennifer A. O'Hern** [1¤a]*, **Adrian Koenen**[2], **Sonja Janson**[1], **Krispin M. Hajkowicz**[3], **Iain K. Robertson**[4], **Sarah E. Kidd**[5], **Robert W. Baird**[1,6], **Steven YC Tong**[7¤b¤c], **Joshua S. Davis**[7¤d], **Phillip Carson**[2], **Bart J. Currie**[1,7], **Anna P. Ralph**[1,7]*

1 Department of Infectious Diseases, Royal Darwin Hospital, Darwin, Australia, 2 Department of General Surgery, Royal Darwin Hospital, Darwin, Australia, 3 Royal Brisbane and Women's Hospital, Brisbane, Australia, 4 College of Health and Medicine, University of Tasmania, Launceston, Tasmania, Australia, 5 National Mycology Reference Centre, SA Pathology, Adelaide, Australia, 6 Territory Pathology, Department of Health, Darwin, Australia, 7 Menzies School of Health Research, Charles Darwin University, Darwin, Australia

¤a Current address: Department of Infectious Diseases, Launceston General Hospital, Launceston, Tasmania, Australia
¤b Current address: Victorian Infectious Diseases Service, The Royal Melbourne Hospital, at the Peter Doherty Institute for Infection and Immunity, Melbourne, Australia
¤c Current address: Department of Infectious Diseases, The University of Melbourne at the Peter Doherty Institute for Infection and Immunity, Melbourne, Australia
¤d Current address: Infectious Diseases Department, John Hunter Hospital, Newcastle, Australia
* anna.ralph@menzies.edu.au (APR); jen.ohern@gmail.com (JAO)

**Data Availability Statement:** All relevant data are within the paper and its Supporting Information files.

## Abstract

### Background

*Cryptococcus gattii* is a globally endemic pathogen causing disease in apparently immune-competent hosts. We describe a 22-year cohort study from Australia's Northern Territory to evaluate trends in epidemiology and management, and outcome predictors.

### Methods

A retrospective cohort study of all *C. gattii* infections at the northern Australian referral hospital 1996–2018 was conducted. Cases were defined as confirmed (culture-positive) or probable. Demographic, clinical and outcome data were extracted from medical records.

### Results

45 individuals with *C. gattii* infection were included: 44 Aboriginal Australians; 35 with confirmed infection; none HIV positive out of 38 tested. Multifocal disease (pulmonary and central nervous system) occurred in 20/45 (44%). Nine people (20%) died within 12 months of diagnosis, five attributed directly to *C. gattii*. Significant residual disability was evident in 4/36 (11%) survivors. Predictors of mortality included: treatment before the year 2002 (4/11 versus 1/34); interruption to induction therapy (2/8 versus 3/37) and end-stage kidney disease (2/5 versus 3/40). Prolonged antifungal therapy was the standard approach in this cohort, with median treatment duration being 425 days (IQR 166–715). Ten individuals had adjunctive lung resection surgery for large pulmonary cryptococcomas (median diameter

**Funding:** The authors received no specific funding for this research.

**Competing interests:** I have read the journal's policy and the authors of this manuscript have the following competing interests:Dr. KH has received advisory board fees and grant support from Gilead Sciences. The authors have no other conflicts to declare.

6cm [range 2.2-10cm], versus 2.8cm [1.2-9cm] in those managed non-operatively). One died post-operatively, and 7 had thoracic surgical complications, but ultimately 9/10 (90%) treated surgically were cured compared with 10/15 (67%) who did not have lung surgery. Four patients were diagnosed with immune reconstitution inflammatory syndrome which was associated with age <40 years, brain cryptococcomas, high cerebrospinal fluid pressure, and serum cryptococcal antigen titre >1:512.

### Conclusion

*C. gattii* infection remains a challenging condition but treatment outcomes have significantly improved over 2 decades, with eradication of infection the norm. Adjunctive surgery for the management of bulky pulmonary *C. gattii* infection appears to increase the likelihood of durable cure and likely reduces the required duration of antifungal therapy.

### Author summary

*Cryptococcus gattii* is an environmental fungus responsible for invasive infection, predominately in the central nervous system (CNS) and lungs. There is little evidence to guide its management. We have found First Nations Australians in this region have one of the highest incidences of *Cryptococcus gattii* infection in the world, with rates possibly increasing.

Mortality was associated with end stage kidney disease, diagnosis in earlier years of the study, and unplanned interruptions to intravenous treatment. Whilst mortality improved through the study, neurological disability such as visual or hearing impairment following cure continues to be seen and is associated with brain cryptococcomas and high cerebral spinal fluid (CSF) pressure at diagnosis or during treatment.

Blocked CSF shunts are uncommon and concerns of such potential complications should not preclude necessary surgical intervention for persisting increased CSF opening pressures or hydrocephalus which has been associated with poor outcome.

Infection recurrence or persistence was associated with large pulmonary cryptococcomas (over 3cm diameter) not undergoing pulmonary surgery and we recommend that surgery be considered for any pulmonary cryptococcoma with a diameter over 2 to 3cm.

Eradication of infection can be expected provided there is adequate therapy but may require very prolonged antifungal treatment.

### Introduction

*Cryptococcus gattii* is an environmental fungus responsible for invasive infection, predominately in the central nervous system (CNS) and lungs. Unlike *C. neoformans*, infection is recognised to commonly occur in apparently immune-competent individuals.

Guidelines and recommendations for management such as those published by the Infectious Diseases Society of America are based on extrapolation from trials in *C. neoformans* infections, individual case reports and series, and expert opinion [1–3]. Guidelines lack precision on selection and duration of antifungal therapy, indications for surgical resection of bulky disease especially in the lung, and indications for CNS surgery to manage high intracranial pressure.

*C. gattii* is endemic in Australia [4], with particularly high incidence in the Northern Territory and amongst First Nations peoples (Aboriginal Australians) [4–6]. A 2000–2007 Australia-wide case series included patients from the Northern Territory [4] but there has been no recent description of the epidemiology, management, and outcomes of *C. gattii* in this highly endemic region.

This study aims to provide a comprehensive, longitudinal description of the epidemiology, management and outcomes for *C. gattii* in an endemic region. We explore factors associated with mortality, immune reconstitution inflammatory syndrome (IRIS), neurological sequelae and persistence or relapse. We also aim to make observations and recommendations regarding the role of surgery in the management of *C. gattii*.

## Methods

### Ethics statement

Ethics approval was provided by the Human Research Ethics Committee of the NT Department of Health and Menzies School of Health Research: HREC-2016-2688. Individual patient consent was waived because only retrospective data collected during routine clinical care were extracted.

We conducted a retrospective cohort study of infection with C. gattii managed at Royal Darwin Hospital, the tertiary referral hospital in the Northern Territory (NT), from January 1996 until September 2018.

### Setting

Royal Darwin Hospital (RDH) is a 360-bed tertiary referral centre that serves a population of 170,000 over an area of approximately 500,000 km$^2$. This region is referred to as Top End, NT (Fig 1). Approximately 20% of the population of this region are Aboriginal Australians. [7].

### Study population

All patients identified with infection due to *C. gattii* between January 1996 and September 2018 were eligible for inclusion. Cases were identified through the hospital Infectious Diseases Department clinical database, microbiology database of cryptococcal antigen (CrAg) results and specimens culture positive for *Cryptococcus* spp, and hospital discharge summary coding data by searching International Classification of Diseases (ICD) codes for cerebral cryptococcosis, pulmonary cryptococcosis and disseminated cryptococcosis (B45.1, 45.0 and 45.7).

Categorisation of cases as confirmed or probable was in accordance with previous definitions [8–10]: individuals with a compatible clinical picture plus positive culture for *C. gattii* were categorised as definite *C. gattii* cases. Those with a compatible clinical picture plus positive histology or cryptococcal antigen but without speciation were included as probable *C. gattii* cases provided they were not known to be HIV positive and not taking immunosuppressive medications. Other immunosuppressive disorders (chronic kidney disease and end stage kidney disease, diabetes, recent or current pregnancy, connective tissue disease, solid organ malignancy, splenectomy, HTLV-1 infection, hazardous alcohol consumption) were permitted. End stage kidney disease was defined as estimated glomerular filtration rate <15mL/min or requiring renal replacement therapy.

11 people included in other case series descriptions [1,4,5,11,12] were also included in this study.

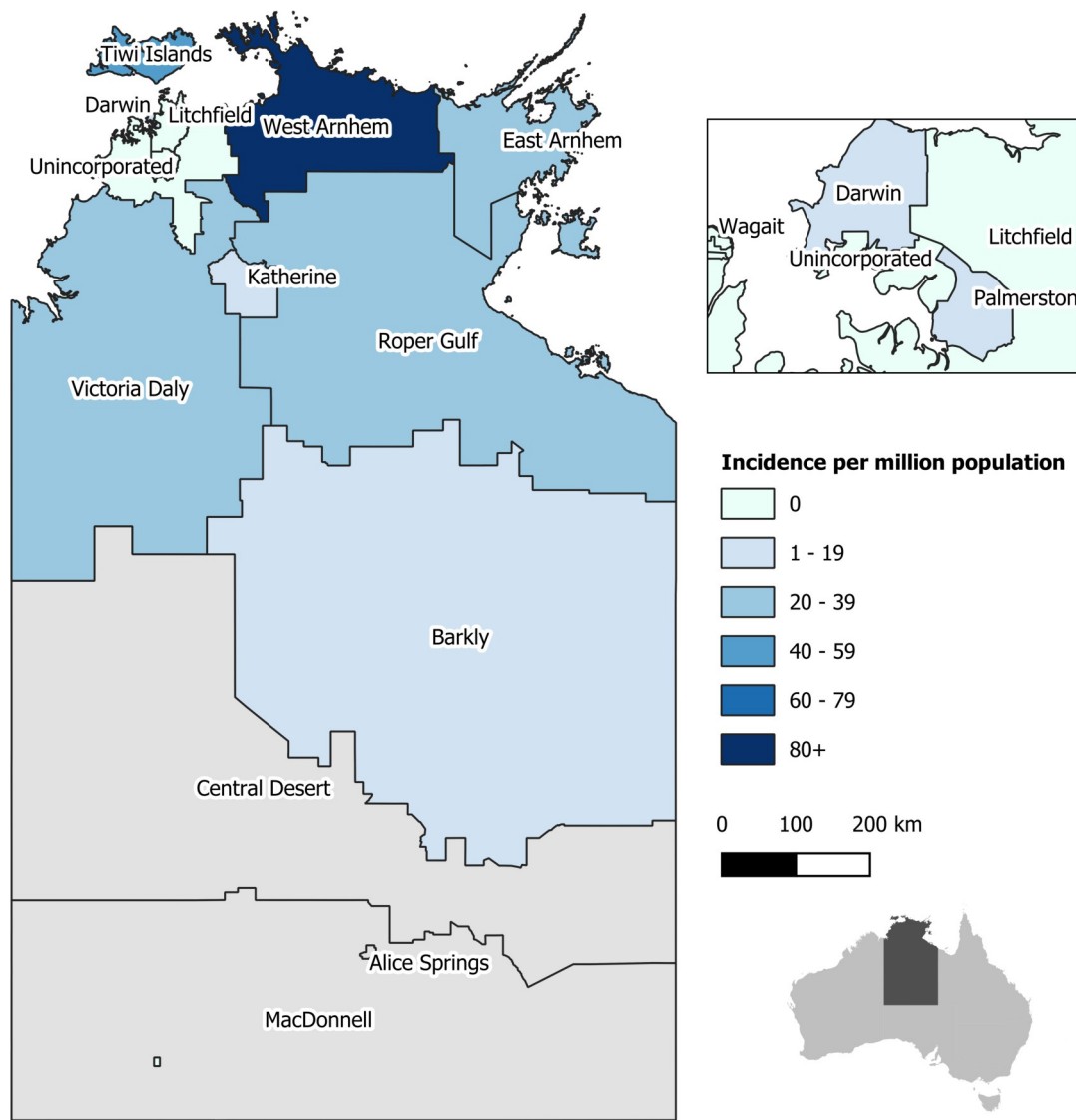

**Fig 1. Northern Territory Top End[a] regional incidence based on available data.** The Northern Territory Top End is defined for for this study to include those regions shaded in blue with exception to Barkly. These regions all feed to Royal Darwin Hospital as their major hospital. (Central Desert, Alice Springs and MacDonnell are not part of this region. Barkly feeds both to RDH and to Alice Springs and subsequently Fig 1 may underestimate the true incidence for that region.) Figure developed using geographic boundary data from the Australian Bureau of Statistics available: https://www.abs.gov.au/AUSSTATS/abs@. nsf/DetailsPage/1259.0.30.0012007.

## Data collection

Demographic, clinical, treatment, and outcome data were extracted from paper and electronic medical records and death certificates, where relevant, using a paper-based case record form and managed using REDCap electronic data capture tools hosted at Menzies School of Health Research [13,14].

Microbiological data were collected from patient notes, the laboratory database and the National Mycology Reference Centre.

Final data collection for outcomes was completed in September 2022.

## Definitions

The geographical location of likely exposure was determined from patients' main residence in the twelve months prior to symptom onset, or prior to diagnosis if patient asymptomatic. Where patients reported equivalent periods of residence in ≥2 locations, attribution was made to the more remote or non-urban region.

Date of diagnosis was defined as the date the first test result confirming diagnosis was reported and treating team impression was of *Cryptococcal* infection.

Anatomical foci of infection were determined from clinical records as documented by treating teams combined with microbiological and radiological findings.

Pulmonary cryptococcoma size was defined as the largest dimension in the radiologist's report made at the time of imaging. High-burden lung disease was defined post-hoc as consolidation of >3 lobes, or cryptococcoma with largest dimension >3cm (in keeping with prior radiological definitions for lung *masses*, versus lung *nodules* with diameter ≤3 cm) [15,16].

Induction antifungal treatment was defined as daily (or equivalent if adjusted for renal function), intravenous therapy with conventional or lipid-based formulation amphotericin, with or without flucytosine, for at least one week. Eradication treatment was defined as oral antifungal therapy, or intravenous amphotericin formulation three times per week (unless adjusted for renal function).

Interruption to induction treatment was defined as no anti-cryptococcal antifungal therapy for more than 96 hours (4 days) either due to unplanned patient discharge or withholding therapy by medical team for example due to suspected antifungal adverse effects.

CSF opening pressure was recorded as 35 cmH$_2$0 for the purpose of statistical calculations if it was documented as '>34 cmH$_2$0' or 'high', where 'high' indicated pressure to or beyond the top of the manometer.

Immune Reconstitution Inflammatory Syndrome (IRIS) was defined as being present if it was diagnosed and documented by the treating team, *and* a clinical and/or radiological deterioration was evident in a previously-improving patient receiving antifungal therapy, and cultures from relevant specimens were negative. This aligns with a previously used cryptococcosis IRIS definition [1].

Cure was defined as cessation of antifungals with no documented relapse by the end of the study. Relapse was defined as clinical evidence of *C. gattii* infection, with microbiological and/or radiological evidence, after completion of planned treatment. Persistence was defined as the impression of persistent cryptococcal infection documented by the treating team with microbiological and/or radiological evidence of ongoing disease during treatment as at September 2022.

Significant disability was defined as a disability attributed to *C. gattii* infection that impaired the patient's ability to function independently or perform their usual activities.

Cause of death was determined by review of death certificates and patient notes.

Outcomes were followed until the most recent documented clinical review for any reason that provided sufficient information to infer outcome from the *C. gattii* infection.

## Cryptococcal antigen testing, Species identification and antifungal susceptibility testing

Cryptococcal antigen titre was measured using the Cryptococcal Antigen Latex Test (Remel USA).

Organism identification to genus level occurred at the Royal Darwin Hospital Pathology Department. Identified *Cryptococcus* isolates were sent to the National Mycology Reference Centre, South Australia, for speciation and susceptibility testing. Until 2014, ID32C

biochemical kits (Biomerieux) were used in combination with L-Canavanine glycine bromothymol blue (CGB) media for identification. From 2014, identification was primarily with Bruker Matrix-Assisted Laser Desorption/Ionization Time-Of-Flight (MALDI-TOF), with CGB media or Internal Transcribed Spacers (ITS) sequencing used if needed.

Susceptibility testing utilised Sensititre YeastOne YO10. In 2013, there was a change in the RPMI (Roswell Park Memorial Institute) broth used with these tests from one that contained 2% glucose to one that contained 0.2% glucose. A verification study at the time found no significant differences to minimum inhibitory concentrations (MICs) for most antifungals apart from an occasional 1- to 2-fold difference in the amphotericin B MIC, either up or down.

### Data analysis

The effect of predictor variables on mortality due to *C. gattii* was estimated as the incidence rate ratio (IRR; 95% confidence intervals; P-values) using multivariable exact Poisson regression. Covariates included in the final model were selected by backward stepwise regression. Analyses were performed using Stata MP2 16.1 (StataCorp LLC, College Station, Tx USA).

Annual incidence rates were calculated on population estimates for each region of Top End NT from Australian census data published every 5 years [7]. Overall incidence for the full duration of the study was calculated by the mean annual incidence rate. (Not all regions had estimates available for the first 4 years of the study. Where this was the case, data from the next available census was used for the purposes of calculating overall incidence.)

## Results

73 possible cases of cryptococcal infection were identified, 45 of whom fulfilled criteria for confirmed (n = 35) or probable (n = 10) *C. gattii* infection and had records available for review (Fig 2 and Tables 1 and S1). All except one were Aboriginal Australians, predominantly from the Arnhem regions of the Northern Territory (Fig 1 and Table 1). Median age was 41 years (range 5–60), four were <18 years (Tables 1 and S1). Median duration of follow up for survivors was 5 years. Multifocal disease was most common (20) followed by pulmonary disease alone (16) (with negative lumbar puncture with or without CNS imaging in 14/16), central nervous system (CNS) disease alone (8) (with negative pulmonary imaging confirmed in 7/8) and fungaemia without identifiable focus (1). 0–4 new diagnoses were made annually (Table 2 and Fig 3), with the highest number of diagnoses made in the last 5 years of the study.

Sixteen individuals had a form of immunosuppression such as diabetes, renal replacement therapy, malignancy or HTLV-1 seropositivity. Of 38 patients tested, none were HIV positive. For 14 individuals, the only documented comorbidity was smoking and/or hazardous alcohol consumption, and 3 people had no documented comorbidities (Tables 1 and S2).

For Top End NT, there was an overall population incidence of 12.9/1,000,000/year for all cases and 9.8/1,000,000/year for confirmed cases. The rate in Aboriginal Australians was 58.9/1,000,000/year for confirmed and probable cases (Table 2).

### Clinical characteristics

Treatment delays were evident with commencement of treatment occurring a median of 26 days [IQR 11,62] after reported symptom onset, with the longest time between onset of headache and presentation with cryptococcal meningitis being 12 months (S1 Table). Time to diagnosis and treatment from presentation to hospital was a median of 3 days [IQR 1,9]. Commonest presenting symptoms were headache and fever in those with CNS infection, and cough and fever in pulmonary infection. 5/28 with CNS disease had no CNS symptoms at presentation with diagnosis of CNS infection by lumbar puncture done for standard workup of *C.*

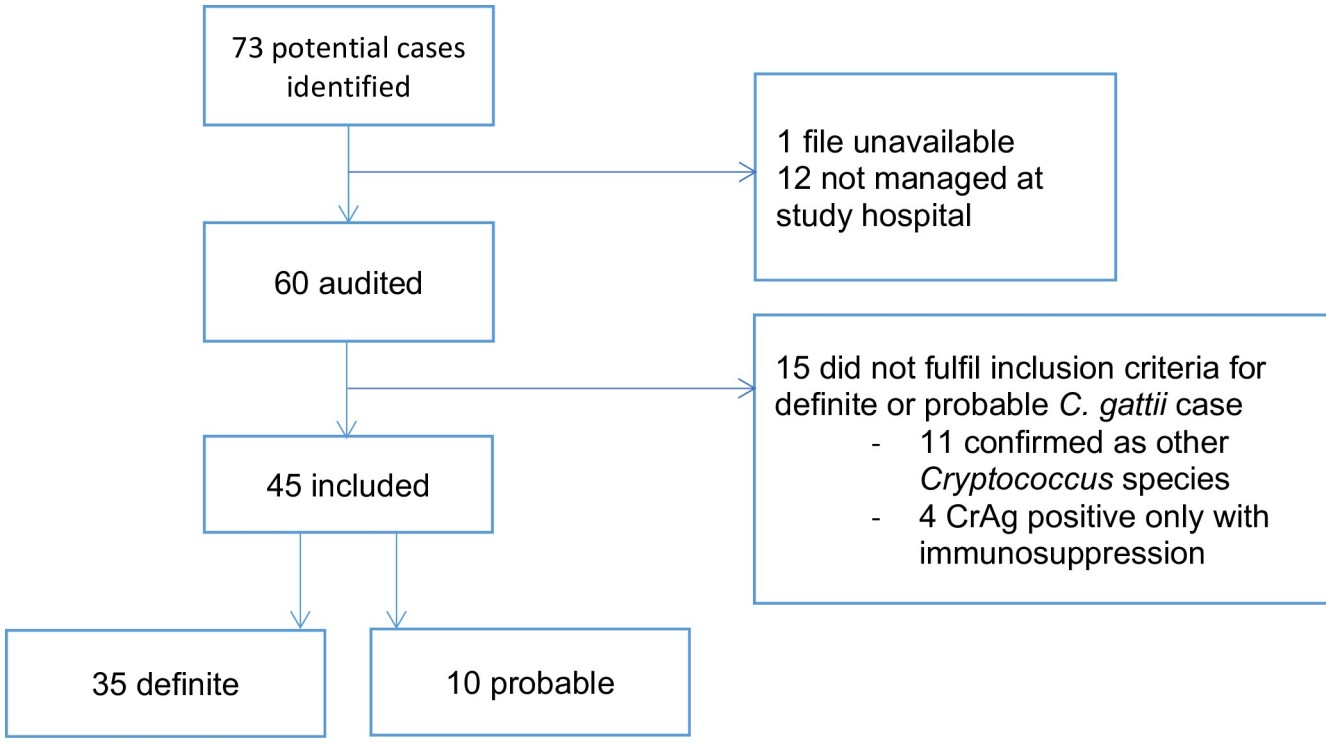

CrAg cryptococcal antigen

**Fig 2. Study diagram.** Note 12 patients not managed at study hospital–these are cases diagnosed at a hospital outside the Top End Region that refers some microbiological specimens to RDH for processing.

*gattii* infection. 4/36 with pulmonary infection had no pulmonary symptoms with diagnosis by pulmonary imaging done for routine workup of *C. gattii* infection (Table 1).

### Antifungal susceptibility testing and treatment

Results from susceptibility testing were available in 25 cases (Table 3). The median fluconazole MIC was 8 μg/mL. 11/25 patients had *C. gattii* isolates with a fluconazole MIC ≥16 μg/mL including 7/13 diagnosed in 2015 or later (Fig 3).

Treatment information was available for 44 patients. All received induction therapy with systemic amphotericin (12 conventional, 25 lipid formulation, 7 both in series) plus oral flucytosine (Table 4). Flucytosine was replaced with an azole antifungal in 6 people due to adverse drug events. Liposomal amphotericin was dosed at 3-4mg/kg/day with the exception of 3 patients prior to 2002 who received lower doses and 3 patients who received higher doses of 4-6mg/kg. Median duration of induction therapy was 42 days. Two patients with pulmonary only disease were cured with eight and nine days induction therapy followed by azole therapy (143 and 99 days respectively).

12 people received four or less weeks induction therapy. In three this was because death intervened prior to four weeks. Of the remaining nine (five pulmonary, two pulmonary plus meningitis, one fungaemia, one meningitis), eight were cured including one who received no oral stepdown therapy. One was subsequently diagnosed with relapse.

**Table 1. Patient characteristics.**

| | All | CNS disease only | Pulmonary disease only | Fungaemia only | Combined CNS and pulmonary disease |
|---|---|---|---|---|---|
| **Number (%)** | 45 (100) | 8 (18) | 16 (36) | 1 (2) | 20 (44) |
| **Age in years: median (range)** | 41 (5–60) | 45.5 (13–57) | 41 (17–60) | 41 | 38.5 (5–59) |
| **Male: No (%)** | 23/45 (51) | 4/8 (50) | 7/16 (44) | 0 | 12/20 (60) |
| **Ethnicity: No (%)** | | | | | |
| **Australian Aboriginal** | 44/45 (98) | 8/8 (100) | 16/16 (100) | 1/1 (100) | 19/20 (95) |
| **Other** | 1/45 (2) | 0 | 0 | 0 | 1/20 (5) |
| **Comorbidity: No (%)** | | | | | |
| ESKD | 5/45 (11) | 1/8 (12.5) | 4/16 (25) | 0/1 | 0/10 |
| Comorbidities assoc with immunosuppression[a] | 16/45 (36) | 3/8 (38) | 7/16 (44) | 0/1 | 6/20 (30) |
| Hazardous alcohol use | 16/45 (36) | 3 /8 (38) | 2/16 (13) | 1/1 (100) | 10/20 (50) |
| Smoking | 26/45 (58) | 6/8 (75) | 11/16 (69) | 1/1 (100) | 8/20 (40) |
| No comorbidity | 3/45 (7) | 1/8 (13) | 1/16 (6) | 0/1 | 1/20 (5) |
| Smoking and/or hazardous alcohol use only comorbidity | 14/45 (31) | 3/8 (38) | 4/16 (25) | 0/1 | 7/20 (35) |
| **Symptom duration before presentation in days: median (range)** | 26 (1–391) | 47 (11–391) | 22 (5–366) | 5 | 21.5 (1–391) |
| **Interruption to induction treatment** | 8/45 (18) | 1/8 (13) | 3/16 (19) | 1/1 (100) | 3/20 (15) |
| **Symptoms: No (%)** | | | | | |
| Fever | 21/45 (47) | 5/8 (63) | 6/16 (38) | 1/1 | 9 /20 (45) |
| Weight loss | 6/45 (13) | 1/8 (13) | 1/16 (6) | 0/1 | 4 /20 (20) |
| Hypotension | 6/45 (13) | 1/8 (13) | 0/16 | 0/1 | 5/20 (25) |
| CNS symptoms[b] | 23/45 (51) | 7/8 (88) | 2/16[c] (13) | 1/1 (100) | 13/20 (65) |
| Pulmonary symptoms[d] | 23/45 (51) | 0/8 | 12/16 (75) | 0/1 | 11/20 (55) |
| **Diagnosis: No (%)** | | | | | |
| Culture positive | 35/45 (78) | 5/8 (63) | 11/16 (69) | 1/1 (100) | 18/20 (90) |
| Culture positive, species not confirmed | 1/45 (2) | 0/8 | 0 | 0/1 | 1/20 (5) |
| Antigen positive, culture negative | 8/45 (18) | 3/8 (38) | 4/16 (25) | 0/1 | 1/20 (5) |
| Histology positive, culture negative | 1/45 (2) | 0/8 | 1/16 (6.3) | 0/1 | 0/20 |

ESKD end stage kidney disease: eGFR<15mL/min or on chronic kidney disease on dialysis

[a] chronic kidney disease on dialysis or ESKD eGFR<15mL/min, diabetes mellitus, pregnancy within 3 months prior to or at diagnosis, connective tissue disease, haematological malignancy, solid organ malignancy, other immunosuppressive disorder, immunosuppressive medication

[b] headache, decreased consciousness, confusion, behaviour change, blurred vision, nuchal rigidity, ataxia, papilloedema, focal neurology.

[c] lumbar puncture and CNS imaging (CT or MRI) both normal

[d] dry cough, productive cough, haemoptysis, pleuritic chest pain, dyspnoea

Three treated prior to 2002 additionally received intraventricular amphotericin via an Ommaya access reservoir (Fig 3).

Eradication therapy was commenced with fluconazole (usually 400mg/day; 800mg/day in 6 patients usually due to high fluconazole MIC) in 35/39 patients who completed induction

**Table 2.** *Cryptococcus gattii* **case numbers and estimated incidence over time for Top End NT[a].**

| Years | All new diagnoses | Culture positive diagnoses | Rate/1,000,000 total population/year[a] | | Rate/ 1,000,000 Aboriginal population/year[a] | |
|---|---|---|---|---|---|---|
| | | | all | culture pos | all | culture pos |
| 1996–1999[a] | 6 | 6 | | | | |
| 2000–2004 | 8 | 7 | 11.2 | 10.1 | 47.4 | 40.6 |
| 2005–2009 | 8 | 5 | 12.9 | 8 | 51.3 | 32 |
| 2010–2014 | 8 | 6 | 10.4 | 7.8 | 52.1 | 39 |
| 2015–2018 | 15 | 12 | 19.1 | 14.2 | 99.6 | 76.6 |
| **Total[a]** | **45** | **35** | **12.9** | **9.8** | **58.9** | **44.8** |

[a] Not all regions had estimates available for the first 4 years of the study. Where this was the case, data from the next available census was used for the purposes of calculating incidence of the entire study.

treatment. Seven patients were changed from fluconazole to alternative agents (itraconazole: two, both in 2002; voriconazole: five, all 2015 or later) for reasons of toxicity, treatment failure or high fluconazole MIC (S3 Table and Fig 3). One patient was commenced on voriconazole when diagnosed with infection persistence 12 months post cessation of fluconazole (fluconazole MIC of the original isolate was 8 μg/mL).

For 22/30 patients confirmed to have completed treatment and duration was known, the median total treatment duration (initiation and eradication phases) was 425 days (IQR 166–715).

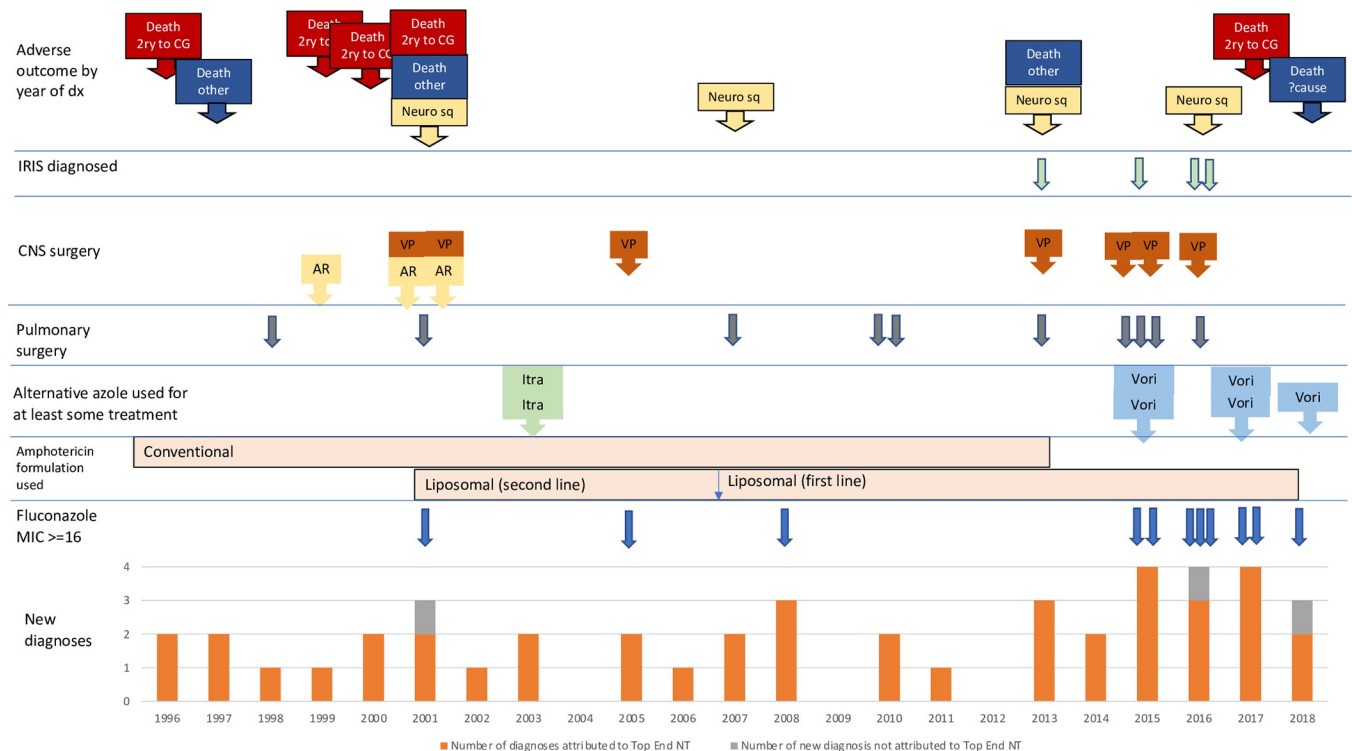

**Fig 3. Timeline.** New diagnoses, management and outcomes of patients infected with *C. gattii* managed at RDH over the 22 years of the study. *Death 2ry to CG* Death attributed to *C. gattii Death Other* Death within 12 months or before completion of treatment attributed to other cause *Neuro sq* Infection cured with significant neurological sequelae *IRIS* immune reconstitution syndrome *CNS* central nervous system *VP* VP shunt inserted *AR* Access reservoir inserted *Itra* itraconazole *Vori* voriconazole.

**Table 3. Antifungal susceptibility.**

| | Number Tested | MIC (µg/mL) | | | | | | | | | | | |
|---|---|---|---|---|---|---|---|---|---|---|---|---|---|
| | | 0.015 | 0.03 | 0.06 | 0.12 | 0.25 | 0.5 | 1 | 2 | 4 | 8 | 16 | 32 |
| Fluconazole | 25 | | | | | | | 1 | 1 | 3 | 9 | 10 | 1 |
| Voriconazole | 20 | 1 | 0 | 3 | 7 | 7 | 2 | | | | | | |
| Itraconazole | 25 | | | 6 | 8 | 10 | 1 | | | | | | |
| Posaconazole | 18 | 2 | 3 | 9 | 4 | | | | | | | | |
| Amphotericin | 25 | | 1 | 1 | 6 | 3 | 5 | 9 | | | | | |
| Flucytosine | 23 | | | | | 1 | 2 | 10 | 7 | 2 | 1 | | |

There were 18 episodes in 14 patients of suspected adverse reaction to antifungal drugs leading to change or withholding of treatment. (S4 Table).

## Outcomes

Nine people (20%) died within twelve months of diagnosis. Five deaths were directly attributable to *C. gattii*, all within four months of diagnosis (Table 4). 4/36 survivors (11%) had significant residual disability. 23/44 (52%) patients required admission to the intensive care or high dependency unit for management of their *C. gattii* infection. Of 36 patients surviving twelve

**Table 4. Treatment and outcome.**

| | All (45) n (%) | CNS disease only (8) | Lung disease only (16) | Fungaemia only (1) | Combined CNS and lung disease (20) |
|---|---|---|---|---|---|
| **Number still on eradication treatment:** | 2[a] | 0 | 1 (6%) | 0 | 1 (5%) |
| **Initiation phase treatment duration in days:** median (range) | 42 (8–180) | 49 (26–120) | 21 (8–68) 2 don't know | 23 | 47 (14–180) |
| **Total treatment duration in days:** median (range) | 403 (28–870) | 240 (173–407) | 384 (28 - >2yr ongoing) | 293 | 555 (113–870) |
| **Ancillary management and surgery** | | | | | |
| Repeated lumbar puncture for high OP | 8 (18%) | 2 (25%) | 0 | 0 | 6 (30%) |
| Ventricular drain (all went onto VP shunt) | 5 (11%) | 0 | 0 | 0 | 5 (25%) |
| VP shunt | 7 (15%) | 0 | 0 | 0 | 7 (35%) |
| Resection of cryptococcoma (lung) | 10 (22%) | 0 | 0 | 0 | 10 (50%) |
| **Outcome** | | | | | |
| **Died within 12 months** | **9 (20%)** | **1 (13%)** | **5 (31%)** | **0** | **3 (15%)** |
| from *C. gattii* | 5 (11%) | 1 (13%) | 3 (19%) | 0 | 1 (5%) |
| from other cause | 3 (7%) | 0 | 2 (13%) | 0 | 1 (5%) |
| from unknown cause | 1 (2%) | 0 | 0 | 0 | 1 (5%) |
| **Cured** | **32 (71%)** | **5 (63%)** | **10 (63%)** | **1 (100%)** | **16 (80%)** |
| with no or minimal symptoms | 26 (58%) | 5 (63%) | 10 (63%) | 1 (100%) | 10 (55%) |
| with neurological disability | 4 (9%) | 0 | 0 | 0 | 4 (20%) |
| with cough, pleuritic pain | 1 (2%) | 0 | 0 | 0 | 1 (5%) |
| after relapse, no or minimal symptoms, | 1 (2%) | 0 | 0 | 0 | 1 (5%) |
| **Persistence, continues on treatment, minimal symptoms** | **2[a] (4%)** | **0** | **1 (6%)** | **0** | **1 (5%)** |
| **Lost to follow up (but expected to be off treatment with no record of death)** | **2 (4%)** | **2** | **0** | **0** | **0** |

[a] 1 recommenced after diagnosis with persistence or recurrence

OP Opening pressure

fc flucytosine

AE Adverse Event

**Table 5. Associations between clinical and laboratory variables and death attributed to *Cryptococcus gattii* disease (multivariable model).**

| | | Mortality from *C. gattii* | Survived | Incidence rate ratio | 95%CI | P-value |
|---|---|---|---|---|---|---|
| Diagnosed 2002–2018 | n (%) | 1/34 (3%) | 33/34 (97%) | 1.00 | | |
| Diagnosed 1996–2001 | n (%) | 4/11 (36%) | 7/11 (64%) | 89.1 | (7.96, 4698) | 0.0001 |
| No interruption to induction treatment | n (%) | 3/37 (8%) | 34/37 (92%) | 1.00 | | |
| Induction treatment interruption | n (%) | 2/8 (25%) | 6/8 (75%) | 62.8 | (4.61, +∞) | 0.0031 |
| Without ESKD | n (%) | 3/40 (7.5%) | 37/40 (92.5%) | | | |
| ESKD | n (%) | 2/5 (40.0%) | 3/5 (60%) | 59.7 | (4.64, +∞) | 0.0030 |
| Symptom (days) before diagnosis | Mean (SD; N) | 21.0 (14.2; 3) | 56.0 (85.7; 36) | 1.27 | (0.60, 2.70) | 0.53 |
| Meningitis | n(%) | 2/27 (7.4%) | 3/18 (17%) | 0.34 | (0.02, 4.13) | 0.55 |
| Lung cryptococcoma | n(%) | 4/36 (11.1%) | 1/9 (11%) | 3.85 | (0.15,303) | 0.67 |
| Cryptococcal antigen titre (log-transformed z-score) | Geometric mean (N) | 1:256 (4) | 1:307 (3) | 1.82 | (0.43, 7.81) | 0.42 |

(IRR and 95% confidence intervals/P values estimated using multivariate exact Poisson regression)

ESKD End stage kidney disease

Note all cases of mortality attributed to *C.gattii* in this study occurred within 12 months of diagnosis.

months or more, 32 were confirmed to have completed treatment and were considered cured, two remain on treatment for extensive inoperable pulmonary cryptococcal disease and two were lost to follow up, though were asymptomatic at last review and remain alive according to NT records and are considered cured.

Factors associated with death attributed to *C. gattii* in multivariable analysis included end stage kidney disease (p = 0.0030), unplanned interruptions to intravenous treatment (p = 0.0031), and diagnosis prior to 2002 (p = 0.0001) (Table 5). Death was not associated with age, cryptococcal antigen titre, site of infection nor symptom duration prior to presentation (Tables 5 and S5). After 2002, only one death was attributed to *C. gattii*: an individual with multiple co-morbidities for whom palliative care was implemented.

Two patients had disease relapse following treatment completion, both with pulmonary infection and likely suboptimal adherence to eradication therapy. Both cases were asymptomatic or minimally symptomatic at the time of diagnosis with recurrence, and were reinitiated on oral azole therapy for 2 years or lifelong respectively. One other patient, having received >5 years treatment to date, continues on voriconazole with an inoperable mediastinal cryptococcal mass encasing the large vessels, but with mass size approximately halved after two years of continuous voriconazole therapy (fluconazole MIC 16μg/mL).

Severe, persisting disability attributed to the *C. gattii* infection occurred in 4/45 (9%) patients comprising visual impairment (3), mobility impairment (2), hearing impairment (1) and dysphasia (1). All had had concomitant CNS and pulmonary infection, with brain cryptococcomas on imaging and opening pressure ≥35cmH$_2$O or hydrocephalus. IRIS was also associated with residual morbidity in 2/4 patients. (S6A and S6B Table).

Of 38 patients who survived to hospital discharge, median duration of admission was 44 days (range 4–280 days).

5/44 (11%) patients were cured with ≤6 months total treatment. Two with pulmonary-only disease received only 8 and 9 days of intravenous induction therapy respectively then prolonged oral azole therapy, and achieved cure. 12 received <4 weeks intravenous induction therapy, with 8 achieving cure and 1 diagnosed with relapse (in 3/12, death intervened prior to 4 weeks; the remaining 9 had all forms of infection: 5 pulmonary, 2 pulmonary plus meningitis, 1 fungaemia, 1 meningitis)(S3 Table).

Of the 11 patients with fluconazole MIC ≥16 µg/mL, 5 were cured with fluconazole as the oral eradication agent after receiving between 14 and 180 days of amphotericin with flucytosine (S3 Table). The others were treated with alternative agents in response to the observed high fluconazole MIC, due to fluconazole adverse reactions or did not receive eradication therapy (S3 Table).

## CNS disease

Of patients with CNS infection, 27/28 had meningitis of whom 18 had lesions consistent with CNS cryptococcomas on imaging and 1/28 had a solitary enhancing lesion on brain MRI only without evidence of meningitis.

23/27 with meningitis had diagnostic lumbar puncture (LP); median opening pressure 23 cmH$_2$0 (range 3–60) (S7 Table). The volume of therapeutic CSF drainage was infrequently documented and highly variable relative to opening pressure.

Nine patients underwent neurosurgery, seven for management of elevated intracranial pressure (5 with opening pressure ≥35cmH$_2$O (no hydrocephalus on imaging), 2 with hydrocephalus on imaging and consequently no lumbar puncture done). Three patients had access reservoir insertion for administration of intraventricular amphotericin, the last in late 2001 (Fig 3 and S6 Table). Complications of CNS surgery included ventriculitis (3), shunt blockage requiring revision (1), and hygroma (1). All seven patients with neurosurgical management of elevated intracranial pressure ultimately had a ventriculoperitoneal (VP) shunt insertion. 7/7 had CSF protein above normal range (>0.45mg/mL) recorded at time of shunt insertion or shortly after. Shunt blockage occurred in 1/3 patients with CSF protein >10mg/mL, and 0/4 with CSF protein <10mg/mL. 2/9 patients having neurosurgery died secondary to *C. gattii* infection, the other 7 were cured, 3 with significant focal neurological deficit.

Five patients with meningitis and very raised intracranial pressure (opening pressure ≥35cmH$_2$O or hydrocephalus on imaging) did not undergo CNS surgery. One died secondary to *C. gattii* infection, three were cured (one with significant neurological deficit) and one continues on treatment for persistent lung infection with no ongoing CNS symptoms. The two patients with good CNS outcomes each had over 20 therapeutic lumbar punctures. In both these patients, opening pressures ≥35cmH$_2$O and requirement for regular lumbar punctures persisted more than 30 days after commencement of treatment.

Of the 16 other patients with meningitis (highest measured opening pressure 25–34 cmH$_2$O [4], <25cmH$_2$O [7], not stated [4]) fifteen were cured (none with residual neurological disability), and one died within 12 months from another cause.

## Pulmonary infection and surgery

Of 36 patients with lung infection, 25 had cryptococcomas and 11 consolidation only. Of those with consolidation, one died due to non-*C. gattii* cause and the remainder were cured with conservative management. Of those with pulmonary cryptococcoma, 10 had surgical resection (pulmonary wedge resection or lobectomy). Generally, those offered surgery had more bulky disease with a maximum cryptococcoma diameter in the 10 operative patients being 6cm (median, range 2.2-10cm), versus 2.8cm (range 1.2-9cm), in those managed non-operatively (Fig 4). Five patients were considered for surgery but did not proceed due to their high anticipated operative risk.

Nine of 10 patients managed with thoracic surgical resection were cured; one died, attributed to *C. gattii* infection. No patient who underwent pulmonary surgery was diagnosed with recurrence or infection persistence post cessation of antifungal therapy. The decision to operate was generally due to anticipation of failure of antifungal therapy, supported in some cases by poor radiological response after weeks to months of antifungal therapy. Lung surgery

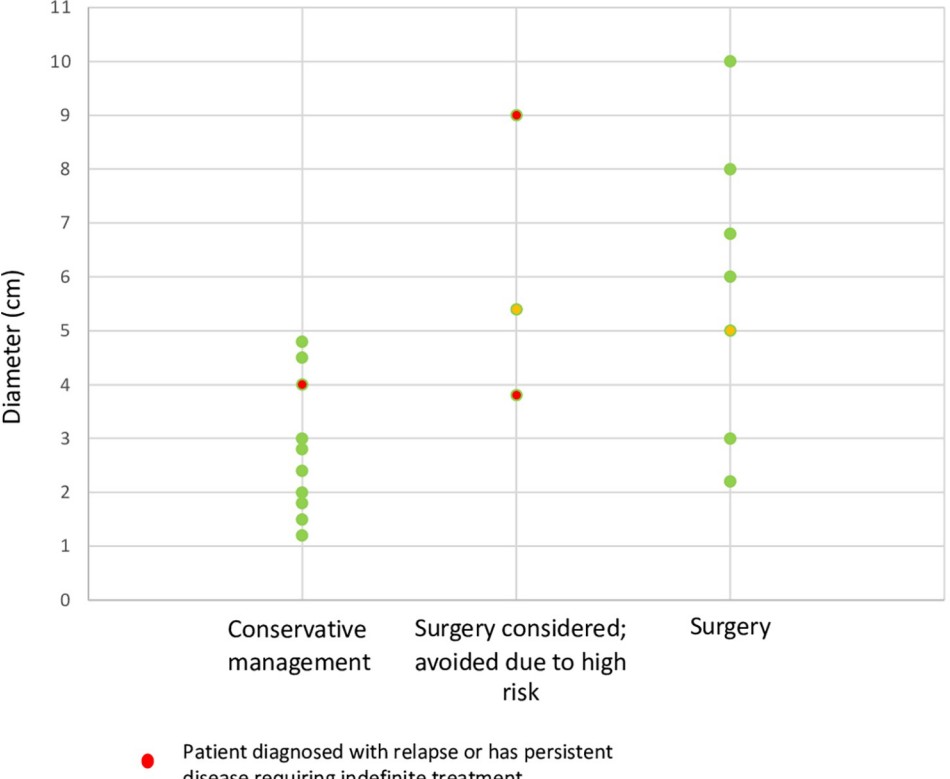

<sup></sup>Size of cryptococcoma not known for 3 patients undergoing surgery (all cured) and 1 not undergoing surgery (died)

**Fig 4. Surgical or conservative management vs diameter of largest pulmonary cryptococcoma and outcome[a].**

occurred after a median of 46 days of antifungal therapy (range 14–287). One or more surgical complications occurred in 7/10 patients and comprised pneumothorax with prolonged air leak (2), prolonged ventilation (1), bleeding (3), arrhythmia (1), empyema (1) and pulmonary embolism with pleural effusion (1).

Of 15 patients with pulmonary cryptococcomas who did not undergo surgery, outcomes following a first course of treatment were: cure (10), recurrence after cessation of therapy (2, 1 now cured, 1 continuing on treatment), and death (3; 1 attributed to *C. gattii*.). Six of these patients managed non-operatively had at least one cryptococcoma >3cm in maximal diameter; their outcomes were cure (2), recurrence (2), persistent infection (1), death–cause not confirmed (1).

Slow radiological improvement was particularly noted in those with cryptococcomas >5cm managed non-operatively or consolidation involving more than three lobes, with no or minimal improvement at 6 weeks in 4/4 cases managed non-operatively who had follow-up imaging available.

## Immune reconstitution inflammatory syndrome

Four patients were diagnosed with IRIS. All were <40 years of age, had long duration of symptoms prior to presentation (median 45 days), were diagnosed after 2012, had brain

cryptococcomas, LP opening pressure >30 cmH$_2$O, and serum CrAg >1:512 and all but one had concomitant pulmonary disease (S5 Table). One had diabetes mellitus; the others had no identified comorbidities. Three had IRIS diagnosed within five weeks of commencing antifungal treatment, one was diagnosed after 14 months. All received and improved with corticosteroids, initially dexamethasone (starting dose 8-16mg or 0.6mg/kg) for 7–48 days before changing to prednisolone to complete a total treatment duration ranging from 63 days to over six months. One had relapses of IRIS on two attempts to wean steroids but responded to reescalation of steroid dose.

## Discussion

This unique case series of *C. gattii* infections in HIV negative individuals spanning two decades reveals changes in medical and surgical management with low mortality for the last 16 years of the study. Just one patient died due to *C.gattii* during this latter time frame which compares favourably with four during the first six years of the study, and mortality reported in other studies [4,5,8]. Changes seen in the second half of the study include shifts away from conventional and intra-ventricular amphotericin, and increased use of voriconazole (though with variable success owing to adverse effects). The last decade of the study also saw a higher number of diagnoses with *C. gattii* infection, increased numbers of patients with an isolate with fluconazole MIC 16µg/mL or higher, emerging recognition and diagnoses of IRIS, and a trend to more frequent pulmonary resections. Our data indicate that this study setting in northern Australia continues to have one of the highest incidences for *C. gattii* infection in the world, supporting previous reports [1,2,4–6,17,18]. This study highlights the substantial morbidity of *C. gattii* infection, including prolonged hospitalization, burdensome treatment in terms of duration and toxicity, and residual disability in nearly 10%.

The cohort in this study were relatively young compared to other studies [8,9,19], likely related to the marked low median age in this region (32 for Northern Territory in the 2016 census) [20]. As seen in other studies for *C. gattii*, unlike *C. neoformans*, most patients lacked clinical evidence of overt immunosuppression [4,9,19]. Recently it has been suggested that *C. gattii* infection may be promoted by defects in the immune system not previously recognized with recent studies suggesting an association with Anti-GM-CSF autoantibodies in some patients with *C. gattii* infection [21,22]. This has yet to be assessed in our cohort.

Knowledge gaps in the environmental and host factors underpinning local disease epidemiology remain unanswered. Every adult in this case series was of Aboriginal ethnicity; the only non-Aboriginal person was a child from a neighbouring Asian country evacuated to Australia for management. To what extent this disparity reflects environmental exposure risk versus host risk factors is unclear. Specific acquired or intrinsic immune pertubations not demonstrable on current routine laboratory may be of importance, but the more relevant issue is to address the burden of chronic conditions related to social determinants of health, which may be playing a role in increasing host susceptibility to infection.

Knowledge gaps in management of *C. gattii* infections that this case series helps to address include support for surgical resection of bulky pulmonary cryptococcomas and emerging experience with different azole antifungals for this indication, most notably voriconazole. Indications for surgery in pulmonary *C. gattii* disease have not been well defined. Infectious Diseases Society of America guidelines recommend considering surgery for pulmonary cryptococcomas if there is compression of vital structures or failure to reduce the size of cryptococcoma after four weeks of therapy, without reference to initial cryptococcoma size [3]. Studies have suggested that surgery may shorten or obviate the need for antifungal treatment [23]. In this case series, we found 0/9 instances of persistence or relapse among survivors of

surgery for pulmonary cryptococcomas, compared with 3/15 instances of relapse and/or persistence in those not undergoing resection, though the groups were not directly comparable, since non-operative management was preferred for more frail patients posing high operative risk. Our centre prefers surgery for bulky disease given local experience and the apparent association with reduction in required antifungal duration and cure, but the level of evidence the current study contributes remains low. We recommend that surgery be considered for any pulmonary cryptococcoma with a diameter over 2 to 3cm. Without surgery, multiple or larger lesions, particularly >5cm, appeared to require prolonged therapy; abbreviated therapy for instance due to patient loss to follow up, could then result in relapse or persistence.

For those with pulmonary infection with consolidation only (no cryptococcoma), other than one patient dying within 12 months of another cause, conservative management resulted universally in cure, though radiological response was often slow.

Outcomes from CNS disease with cryptococcal disease have previously been strongly linked to management of elevated intracranial pressure [3]. The wide variability of CSF drained for different opening pressures suggests there may be a role for written guidance on the volume of CSF to be removed for given opening pressures. IDSA guidelines recommend the pressure be reduced by 50% if it is extremely high or to a normal pressure of <20cm $H_2O$ [3], however a precise guidance to volume that should be removed is not provided and there is no specific mention for recommendation of exit pressure measurement.

The indication for CNS surgery is not precisely defined. It is suggested to be considered for patients requiring daily lumbar punctures [3]. The numbers in our study are small and comparison of outcomes between CNS surgical and non-surgical groups is limited by the fact that those with more severe disease were more likely to undergo surgery. It is notable that in those with opening pressures below 35cm$H_2O$, outcomes were generally good for non-surgical management with repeat lumbar punctures when required, although only three patients in this group had a highest opening pressure 25-34cm$H_2O$. Conversely, in those with opening pressure $\geq$35cm$H_2O$, good outcomes necessitated either surgery or multiple lumbar punctures. It is notable that for the majority of this study, there was no specialist neurosurgeon working at the hospital which may have led to increased management with repeated lumbar punctures with a higher threshold for surgery. Nevertheless, we believe it is important to emphasise the importance of surgery when there is persisting high opening pressure or hydrocephalus and our study showed that blocked or infected CSF shunts are uncommon, and concerns of such potential complications should not preclude necessary surgical intervention.

Traditional guidelines recommend fluconazole for treatment of *C. gattii*. There is little clinical data currently to guide interpretation of antifungal MICs in treatment of cryptococcal infection. The fluconazole MICs seen in this cohort were high compared to other Australian and international studies [1,19,24]. We did not document treatment failures in individuals with high fluconazole MIC treated with fluconazole though we note the predominant use of a higher dose of 400mg/day (compared with 200mg/day recommended for maintenance by Infectious Diseases Society of America) [3]. We did observe clinician preference to use voriconazole later in the study period for *C. gattii* with fluconazole $\geq$16 μg/mL, though its use was often limited by toxicity and in such patients there was usually a good outcome when treatment subsequently reverted to fluconazole. We do note however particular success in one more recent patient in this series with a very large inoperable mediastinal cryptococcal mass, anticipated to need lifelong antifungal therapy, who has had good radiological response and clinical improvement on voriconazole.

Similarly, amphotericin MICs seen particularly in the latter part of this study were relatively high compared to those found in previous studies [25,26]. This did not seem to be associated with poorer outcomes, but cases were heterogeneous with multiple determinants affecting

outcomes. Published guidelines recommend liposomal amphotericin dose (3-4mg/kg) with possible consideration for doses up to 6mg/kg for treatment failure or high burden CNS infection [3], consistent with doses used in this study. In recent years, there has been reported to be no difference in outcomes with lower dose amphotericin [19], however given the amphotericin MICs in our study ($\geq$0.5 µg/mL in 14/25 tested), caution with such an approach to dosing may be warranted.

Patients in our study generally received prolonged induction therapy with an amphotericin formulation (median 6 weeks, minimum 8 days) (Table 4). There is no sound evidence base to inform the safe minimum treatment duration for different forms of *C. gattii* infection. *C. gattii* is recognized to be more complicated to treat than *C. neoformans* [1,3,27]. Australian guidelines from 2013 recommend 2 weeks induction therapy for isolated pulmonary diseases [1] whereas our regional guideline (NT) has to date recommended four weeks minimum induction treatment for all forms of *C. gattii*. This approach contrasts with IDSA guidance and other expert advice support shorter intravenous (induction) courses for both *C. neoformans* and *C. gattii* pulmonary infections in those at lower risk of failure [3, 28] or even use of oral azole therapy only for isolated pulmonary infection. Recent trial findings of success of single high-dose amphotericin and 5-flucytosine induction for *C. neoformans* meningitis in HIV-positive people [29] raise consideration of whether shorter intravenous courses may also be possible for *C. gattii.* In our setting, this needs to be balanced against the challenges of ensuring oral medication supply, adherence and patient follow-up in people returning to very remote areas of Northern Australia. Our data however did show that cure was achievable if <14 days induction therapy was given in pulmonary disease (though numbers were very small, n = 2) and cure was also achievable with <28 days of intravenous induction therapy in all forms of disease but with a relapse rate of 1/9 (11%). Relapse also occurred in one individual despite 6 weeks of intravenous therapy. We conclude that regional guidelines can be revised to recommend that intravenous induction therapy duration for *C. gattii* infection can be tailored to burden of disease, co-morbidities and ability to access follow-up out of hospital care, from two weeks for uncomplicated pulmonary disease to prolonged (potentially months) for extensive, persistently culture positive disease.

We found that chief factors associated with *C. gattii* mortality were patient-related (end-stage kidney disease, unplanned interruptions to induction treatment) and diagnosis prior to 2002, not infection severity measures. We speculate that medical, surgical and intensive care approaches have improved over time accounting for the lower later mortality. Previous studies have also noted an association between mortality and immunocompromise [2,8]. The immune modulation that accompanies end-stage kidney disease [30] may predispose to poorer outcomes in *C. gattii* infection; end-stage kidney disease also limits or complicates available treatment options.

Conversely, neurological morbidity or infection persistence requiring prolonged or indefinite treatment were more related to infection severity. Neurological sequelae were more common in those with elevated opening pressure or hydrocephalus and brain cryptococcomas. Relapse and/or persistent infection requiring ongoing treatment was associated with pulmonary cryptococcomas over 3cm, although medication adherence issues may have also contributed.

The mechanism of IRIS in HIV negative individuals with *C. gattii* is incompletely understood [2]. IRIS is hypothesized to occur in this setting when the transient immunosuppression or immune-evasion, attributed to direct effects of cryptococcal capsule components, resolves with successful treatment, leading to an exaggerated inflammatory response and worsening of clinical and/or radiological findings [31–33]. In our study, the few individuals with IRIS all had CNS disease including brain cryptococcomas, in keeping with previous studies [1,8]. All

also had significant disease burden with elevated markers of severity including opening pressure, and high serum CrAg titres (>1:512), and all but one had concomitant pulmonary disease. To our knowledge, IRIS has not previously been shown to be associated with *C. gattii* disease burden although this association has been described for IRIS in *C. neoformans* infection in HIV positive individuals [1].

The main limitation of this study is that this is a single-centre, small, retrospective study. Data including treatment or outcome details were missing for some individuals. Small numbers limited the ability to draw firm conclusions and the multivariable analysis should be considered exploratory only. There is a chance that some culture-negative cases, while meeting the case definition for *C. gattii*, may have been due to *C. neoformans;* but conversely, the definition may have excluded some culture-negative *C. gattii* cases in immunocompromised patients. Relapse may be underestimated in those diagnosed later in the study.

Strengths of the study include the extended period of observation over 22 years allowing for treatment trends to be tracked over time, as well as long-term outcomes to be documented; and the region serviced by only one major hospital allowing for detailed review of cases.

In conclusion, this case series offer unique insights into management practices and outcomes for *C. gattii* CNS and pulmonary infections in an HIV negative population in a high-burden environment. Whilst a relatively low mortality was seen for the last 15 years of the study, morbidity associated with this infection remains high. Early diagnosis and judicious use of combined surgical and medical approaches are needed. Our long term follow up confirms that with such therapy, cure and eradication of *C. gattii* infection can be expected, in contrast to the uncertainties of cure following therapy of *C. neoformans* infection.

## Supporting information

**S1 Table. Line listing of *Cryptococcus gattii* patients.**
(XLSX)

**S2 Table. Co-morbidities.**
(PDF)

**S3 Table. Treatment and outcomes for individuals with fluconazole MIC ≥16 mg/mL and those who received eradication agents other than fluconazole for at least some of their eradication.**
(PDF)

**S4 Table. Suspected adverse reactions to antifungal drugs requiring change to or withholding of treatment.**
(PDF)

**S5 Table. Predictors of death attributed to *Cryptococcus gattii* in a case series of patients with *Cryptococcus gattii* disease; showing final prediction model plus the individual potential predictors excluded from the final model.**
(PDF)

**S6 Table.** A: Patient and infection characteristics in those with Immune Reconstitution Inflammatory Syndrome (IRIS). B: Neurological disability in those living more than 12 months.
(PDF)

**S7 Table. Highest opening pressure on lumbar puncture for patients with CNS infection.**
(PDF)

## Acknowledgments

We greatly thank Royal Darwin Hospital Infectious Diseases physicians and registrars for providing clinical care; Dr Femy Koratty, Dr Jack Wang and Dr Rosie Stewart for assistance in data collection and Dr Chris Lowbridge for assistance with figure production.

## Author Contributions

**Conceptualization:** Jennifer A. O'Hern, Adrian Koenen, Steven YC Tong, Joshua S. Davis, Phillip Carson, Bart J. Currie, Anna P. Ralph.

**Data curation:** Jennifer A. O'Hern, Krispin M. Hajkowicz, Sarah E. Kidd, Robert W. Baird, Steven YC Tong, Anna P. Ralph.

**Formal analysis:** Jennifer A. O'Hern, Iain K. Robertson, Sarah E. Kidd, Robert W. Baird, Anna P. Ralph.

**Investigation:** Jennifer A. O'Hern, Adrian Koenen, Sonja Janson.

**Methodology:** Jennifer A. O'Hern, Adrian Koenen, Krispin M. Hajkowicz, Steven YC Tong, Phillip Carson, Bart J. Currie.

**Project administration:** Jennifer A. O'Hern.

**Resources:** Anna P. Ralph.

**Software:** Jennifer A. O'Hern.

**Supervision:** Steven YC Tong, Phillip Carson, Bart J. Currie, Anna P. Ralph.

**Visualization:** Jennifer A. O'Hern.

**Writing – original draft:** Jennifer A. O'Hern.

**Writing – review & editing:** Jennifer A. O'Hern, Adrian Koenen, Sonja Janson, Krispin M. Hajkowicz, Iain K. Robertson, Sarah E. Kidd, Robert W. Baird, Steven YC Tong, Joshua S. Davis, Phillip Carson, Bart J. Currie, Anna P. Ralph.

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
