## [Decision Letter · Decision Letter 0]

30 Nov 2022

Dear Dr O'Hern,

Thank you very much for submitting your interesting manuscript "EPIDEMIOLOGY, MANAGEMENT AND OUTCOMES OF CRYPTOCOCCUS GATTII INFECTIONS: A 22-YEAR COHORT" for consideration at PLOS Neglected Tropical Diseases. As with all papers reviewed by the journal, your manuscript was reviewed by members of the editorial board and by two independent reviewers. In light of the reviews (below this email), we would like to invite the resubmission of a significantly-revised version that takes into account the reviewers' comments. 

We cannot make any decision about publication until we have seen the revised manuscript and your response to the reviewers' comments. Your revised manuscript is also likely to be sent to reviewers for further evaluation.

Sincerely,

Joshua Nosanchuk, MD

Section Editor

Joshua Nosanchuk

Section Editor

Reviewer's Responses to Questions

**Key Review Criteria Required for Acceptance?**

**Methods**

-Are the objectives of the study clearly articulated with a clear testable hypothesis stated?

-Is the study design appropriate to address the stated objectives?

-Is the population clearly described and appropriate for the hypothesis being tested?

-Is the sample size sufficient to ensure adequate power to address the hypothesis being tested?

-Were correct statistical analysis used to support conclusions?

-Are there concerns about ethical or regulatory requirements being met?

Reviewer #1: The objective, study design and population are clearly stated and appear appropriate. 

In the abstract, it is stated as “retrospective case series” and in the main text, it is stated “retrospective cohort” study. It seems to be a combination of both. 

Authors provide a comprehensive, longitudinal description of the epidemiology, management, and outcomes of Cg infection. The authors explored factors associated with mortality, immune reconstitution inflammatory syndrome, neurologic sequelae and persistence or relapse. There are no concerns for ethical or regulatory issue. 

Additional comments: 

*Lines 118: It is not clearly stated what the comparison groups were in the univariable analysis. 

*Lines 89; 376. Authors discuss cure and relapse. A limitation to consider is that the authors may have under-estimated relapse due to insufficient follow up time, especially for the cases diagnosed during the latter period. 

*Lines 84-88: 

Authors discuss IRIS in this section. 

It is not entirely clear if IRIS is the correct terminology (as there was no clear “immune reconstitution”, although that could be debatable) or if PIIRS would be the more appropriate terminology. However, there does not appear to be clear consensus on this terminology in general (not referring to this study) and PIIRS terminology was only introduced in 2015. Authors do reference a study on PIIRS (ref 25, line 445) in the discussion section and appear to be interpreting IRIS and PIIRS interchangeably. While this point may not be completely relevant to this study, could be a limitation in accurate classification of IRIS.

Reviewer #2: See below

**Results**

-Does the analysis presented match the analysis plan?

-Are the results clearly and completely presented?

-Are the figures (Tables, Images) of sufficient quality for clarity?

Reviewer #1: Overall, the analysis match the analysis plan and the results are clearly and completely presented. Few comments below: 

*Table 5: 

The small number of events limit the number of variables that can be included in the model. The wide CI likely reflects the high variability of the estimates due to the small sample size. It may help the readers to clarify which variables were included as adjustment variables and consider including few pre-specified variables (that are biologically relevant or confounders) given limited number of events and small sample size. Readers will need to interpret the findings with the limitations in mind. 

*Line 134: “All except one were Aboriginal Australians, predominantly from the Arnhem regions of the Northern Territory (Figure 2)..”

-Authors may have meant to state, “Table 1” instead of “Figure 2” here. 

*Table 1. 

The denominator number is not always included. May be better to be consistent. Could include denominator in all cells. Alternatively, can remove the denominator number but include a total n in the top row and indicate in the footnote if there are few missing. 

*Line 184, authors state “All received induction therapy with systemic amphotericin..” 

Was that the case in patients with pulmonary only disease as well?

*Line 235: Authors state “Death was not associated with age, anatomical site of infection…”

However, the data on association between anatomical site of infection and death is not presented in table 5. 

*Table 1

It may help the readers if the authors give a clearer definition of how “CNS disease only” and “pulmonary disease only” were categorized. It may be of interest given the higher mortality in pulmonary disease compared to CNS disease (as presented in table 4). 

For example, in table 1, two patients with “pulmonary disease only” had CNS symptoms. Would it be correct to assume that the patients had a lumbar puncture and CSF cultures were negative ruling out CNS disease despite the symptoms? 

*In Table 1, authors may consider including the variables they found to be associated with mortality (e.g., ESKD, treatment interruption)

*May be helpful to clarify how “treatment interruption” was defined.

Reviewer #2: See below

**Conclusions**

-Are the conclusions supported by the data presented?

-Are the limitations of analysis clearly described?

-Do the authors discuss how these data can be helpful to advance our understanding of the topic under study?

-Is public health relevance addressed?

Reviewer #1: *It is interesting that disease with Cg was mostly seen in Australian Aboriginal who were relatively young, median age of 41, majority with comorbid conditions. What may be some of the reasons this group of patient population are particularly vulnerable? Or is it more reflective of the general patient population in Northern Australia? Would it be possible other socioeconomic factors may be in play? How generalizable are the findings to Cg cases in other parts of the world?

*May consider elaborating on why smoking and alcohol use were included as comorbid conditions, their relevance with Cg infection. 

*The treatment delay from time of symptom onset is notable. What were the factors that led to the delay? Was it because clinicians did not suspect the disease during the earlier period, or patients present with atypical symptoms or healthcare access issue? Are there ways improvements can be made? 

*The decrease in mortality in the latter period is notable. Could it be that more milder cases were detected in the latter period with increased or improved use of diagnostics?

*It is interesting that pulmonary cryptococcomas is associated with higher morbidity or mortality compared to CNS disease and that symptom duration before presentation was shorter in pulmonary vs CNS disease, which seem to differ from that due to Cn infection in HIV. As the authors present, optimal management/treatment approach between Cg and Cn infection likely differ and need to be studied more. This study would be an important contribution to the existing literature.

Reviewer #2: See below

**Editorial and Data Presentation Modifications?**

Reviewer #1: Minor revision

Reviewer #2: The manuscript (in particular the Results and Discussion sections) is very long and detailed. I would suggest making the report more concise, and moving some parts of the data to the Appendix.

**Summary and General Comments**

Reviewer #1: Authors present an extensive case description of a relatively rare disease over 22 years and make insightful and novel points. The comparison of mortality between early vs. late period is interesting and may reflect changing epidemiology, therapeutics, and/or learning curve with management. The finding of rise in fluconazole MIC over time may have important public health implication and perhaps use of voriconazole or other newer therapeutics need to be studied more. 

This study will make an important addition to the existing body of the literature. The paper also uncovers several important questions that we have little data in with limited guidelines (e.g., monitoring of azole MIC and use of fluc vs vori; surgical vs medical management of cryptococcomas; diagnosis/management of IRIS). The study emphasize the need for better diagnostics and therapeutics. 

Authors report important limitations of their study including single center study and small sample size that limited powered analysis. There was likely residual confounding or confounding by indication when examining outcomes by surgical vs medical approach as the authors discuss.

Reviewer #2: O’Hern and colleagues performed a single center retrospective study looking at the epidemiology, management and outcomes of Cryptococcus gattii infections managed at Royal Darwin Hospital, Northern Territory, Australia. Forty-five cases were included, diagnosed over a 22-year study period (Jan 1996 to Sept 2018). The main study on this question is the Australia-wide retrospective study done by S. Chen and colleagues, in which 86 patients from 17 institutions were included (study period: 2000 – 2007). The study done by O’Hern and colleagues is interesting because their local setting is very particular, with First Nations Australians in this region having a particularly high incidence of C. gattii infection, and specific characteristics. 

Here are my comments and suggestions:

1_ The manuscript (in particular the Results and Discussion sections) is very long and detailed. I would suggest making the report more concise, and moving some parts of the data to the Appendix. 

2_ Definitions used: 

- How was the ‘date of diagnosis’ defined? 

- The authors state that the geographical location of likely exposure was determined from patient’s main residence ‘in the prior 12 months’. Do they refer to the 12 months before diagnosis, or before onset of symptoms? This is important because the time from onset of symptoms to diagnosis is sometimes very long in patients with cryptococcosis. 

- An elevated CSF opening pressure was defined as ≥ 35 cmH20. Please justify the choice of this high threshold. 

3_ Flow chart: a significant proportion of the patients (n=12) were excluded because they were ‘not managed at study hospital’ (Royal Darwin Hospital). Were these patients diagnosed at Royal Darwin Hospital? Systematically excluding this important proportion of the eligible patients may be associated with a high risk of bias, given that these patients possibly had significantly different characteristics, managements, and outcomes. 

4_ Comorbidities: Many patients had few or no major comorbidities. Please discuss that recent studies suggested that C. gattii infections may be promoted by relatively subtle immunocompromising conditions that are not routinely searched for, such as the presence anti-GM-CSF antibodies (Kuo et al, CID 2021)

5_ Incidence of cryptococcosis (Table 2): the local incidence of C. gattii infection has greatly increased in the last period (2015-2018). Could this be explained by a change in diagnostic methods? e.g., was the CrAg latex test replaced by the more sensitive and now widely used LFA? 

6_ Cryptococcus CNS involvement may be asymptomatic. It appears that most patients with CNS involvement presented with neurological symptoms (see Table 1). Please state whether a lumbar puncture (± brain imaging) was systematically done at time of diagnosis of cryptococcosis, even in the absence of neurological symptoms (e.g., in patients presenting with signs of C. gattii lung infection only)

7_ Management: 

a) Patients with C. gattii infection received prolonged induction therapy (i.e., amphotericin-based therapy), with generally at least 6 weeks given for CNS disease, and a median of 3 weeks given for lung disease only (Table 4). This is in line with Australian guidelines for the treatment of cryptococcosis (S. Chen et al, IMJ 2014), but the evidence supporting this recommendation seems to be of particularly low quality. Please comment on this important point. Are there any comparative data on this question? What are the potential advantages and disadvantages of prolonged induction therapy? Given that the recently published AMBITION RCT has shown that C. neoformans meningitis can be treated with single-dose liposomal amphotericin B (combined with flucytosine and fluconazole), could amphotericin-based therapy be shortened in selected patients with C. gattii infection? (e.g., patients with meningitis but no cryptococcoma, or patients with lung infection only)

b) Were antifungal susceptibility testing results reported to treating physicians? If yes, how did that influence antifungal management? 

8_ Outcomes: the authors performed a multivariable analysis to identify factors associated with mortality in patients with C. gattii infection. 

- Please clearly define what outcome variable was used for this analysis. Was it all-cause mortality, or death attributed to C. gattii? Please also confirm that this was at 12 months post-diagnosis of cryptococcosis, not at last follow-up visit.

- The relevance of doing a multivariable analysis appears limited. First, the number of deaths was very low (5 related deaths/45 patients), and prediction models using small data sets lead to very uncertain predictions, as illustrated by the very wide 95% CIs obtained. Second, the clinical relevance of performing a multivariable analysis is limited by the observations that only one death occurred among cases diagnosed after 2002, and that most deaths occurred in patients with lung infection only.

- The variable ‘induction treatment interruption’ was selected as an independent variable, for this multivariable analysis (this was the only management variable used). I would recommend not using this type of variable given the retrospective design of the study, and the high risk of bias associated with these variables.

PLOS authors have the option to publish the peer review history of their article (what does this mean?). If published, this will include your full peer review and any attached files.

Reviewer #1: No

Reviewer #2: No
---

## [Editor Report · Decision Letter 1]

12 Feb 2023

Dear Dr O'Hern,

We are pleased to inform you that your manuscript 'EPIDEMIOLOGY, MANAGEMENT AND OUTCOMES OF CRYPTOCOCCUS GATTII INFECTIONS: A 22-YEAR COHORT' has been provisionally accepted for publication in PLOS Neglected Tropical Diseases. You and your co-authors are to be commended for the thoughtful and thorough revision, which effectively addressed the concerns noted by the reviewers.

Best regards,

Joshua Nosanchuk, MD

Section Editor

---

## [Editor Report · Acceptance letter]

2 Mar 2023

Dear Dr O'Hern,

We are delighted to inform you that your manuscript, "EPIDEMIOLOGY, MANAGEMENT AND OUTCOMES OF CRYPTOCOCCUS GATTII INFECTIONS: A 22-YEAR COHORT," has been formally accepted for publication in PLOS Neglected Tropical Diseases.

Best regards,

Shaden Kamhawi

co-Editor-in-Chief

Paul Brindley

co-Editor-in-Chief
